The impact of an invasive ambrosia beetle on the riparian habitats of the Tijuana River Valley, California

Boland John M. JohnBoland@sbcglobal.net
Southwest Wetlands Interpretive Association , Imperial Beach , CA , United States
Huber Dezene
Electronic publication date: 2016 Jun 21
Publication date: 2016
Volume: 4
Electronic Location ID: e2141
Received 2016 Apr 17; Accepted 2016 May 25
Copyright: ©2016 Boland
Copyright year: 2016
Copyright holder: Boland
License: This is an open access article distributed under the terms of the Creative Commons Attribution License, which permits unrestricted use, distribution, reproduction and adaptation in any medium and for any purpose provided that it is properly attributed. For attribution, the original author(s), title, publication source (PeerJ) and either DOI or URL of the article must be cited.
License URL: https://creativecommons.org/licenses/by/4.0/

Keywords: Euwallacea, Riparian forest, Invasive species, Novel disturbance, Salix gooddingii, Kuroshio Shot Hole Borer, Salix lasiolepis

Funding: The author received no funding for this work.

==============================
The Tijuana River Valley is the first natural habitat in California to be substantially invaded by the Kuroshio Shot Hole Borer (KSHB, Euwallacea sp.), an ambrosia beetle native to Southeast Asia. This paper documents the distribution of the KSHB in the riparian vegetation in the valley and assesses the damage done to the vegetation as of early 2016, approximately six months after the beetle was first observed in the valley. I divided the riparian habitats into 29 survey units so that the vegetation within each unit was relatively homogenous in terms of plant species composition, age and density. From a random point within each unit, I examined approximately 60 individuals of the dominant plant species for evidence of KSHB infestation and evidence of major damage such as limb breakage. In the 22 forested units,I examined the dominant arroyo and black willows (Salix lasiolepis Benth. and S. gooddingii C.R. Ball), and in the seven scrub units, I examined mule fat (Baccharis salicifolia (Ruiz & Pav.) Pers.). Evidence of KSHB infestation was found in 25 of the 29 units. In the forest units, infestation rates ranged from 0 to 100% and were high (>60%) in 16 of the units. In the scrub units, infestation rates ranged from 0 to 33%. Infestation rates were significantly correlated with the wetness of a unit; wetter units had higher infestation rates. Evidence of major physical damage was found in 24 units, and dense stands of willows were reduced to broken trunks in several areas. Overall, I estimated that more than 280,000 (70%) of the willows in the valley were infested, and more than 140,000 had suffered major limb damage. In addition, I recorded evidence of KSHB infestation in the other common plant species in the valley; of the 23 species examined, 14 showed evidence of beetle attack. The four species with the highest rates of infestation were native trees in the Salicaceae family. The three species considered to be the worst invasive plants in the valley, Ricinus communis L., Tamarix ramosissima Ledeb. and Arundo donax L., had low rates of infestation. Several findings from this study have significance for resource managers: (1) the KSHB attack caused extensive mortality of trees soon after being first discovered so, if managers are to control the spread of the beetle, they will need to develop an effective early detection and rapid response program; (2) infestation rates were highest in units that were wet, so resource managers trying to detect the beetle in other areas should thoroughly search trees near water, particularly nutrient-enriched water; (3) the infestation appears to be a novel form of disturbance, and the affected forests may need special management actions in order to recover; and (4) the infestation has altered the structure of the forest canopy, and this is likely to promote the growth of invasive plant species that were relatively inconspicuous in the forests prior to the beetle attack but will now need more attention.

Introduction

Accidentally-introduced insect pests have caused major economic losses and environmental damages within the US (Pimentel, Zuniga & Morrison, 2005). Examples include the elm bark beetle (Scolytus multistriatus Marsham), which has spread Dutch elm disease in North America and killed an estimated 75% of all the elms (Ulmus spp.; Kendrick, 2000), and the balsam woolly adelgid (Adelges piceae (Ratz.)), which has severely damaged the spruce-fir forests of southern Appalachia, killing up to 95% of the Fraser fir trees (Abies fraseri (Pursh) Poir) and substantially changing the avian community (Rabenold et al., 1998; Gandhi & Herms, 2010).

Two accidentally-introduced ambrosia beetles are threatening to cause similar ecosystem-wide damages to the riparian habitats of southern California. These beetles are the Polyphagous Shot Hole Borer (PSHB; Euwallacea sp. near fornicatus; Coleoptera: Curculionidae: Scolytinae) and the Kuroshio Shot Hole Borer (KSHB, Euwallacea sp.; Eskalen, 2016). The two species are morphologically identical and are distinguished by their DNA sequences and by their associated fungi (Eskalen, 2016). They are part of a species complex that also includes the Tea Shot Hole Borer (Euwallacea fornicatus (Eichhoff)). The PSHB was first documented in Los Angeles County in 2003, and the KSHB was first observed in San Diego County in 2012 (Eskalen et al., 2013; Eskalen, 2016; Umeda, Eskalen & Paine, 2016). Both are believed to be native to Southeast Asia and both attack many tree species in southern California, including native species, landscape trees, and the economically important avocado (Persea americana Mill.; Freeman et al., 2013; Eskalen et al., 2013). The ever-increasing list of reproductive host plants used by these species is currently at 41 species for the PSHB and 15 species for the KSHB (Eskalen, 2016).

Both ambrosia beetles were initially observed to cause problems in urban and agricultural settings, but their recent appearance in natural settings has raised grave, new concerns. Both damage or kill trees through their boring activities and their spread of fungal pathogens. Females bore into tree trunks, create networks of tunnels in the xylem, inoculate the tunnels with a fungus (e.g., Fusarium sp.), and live in the tunnels eating the fungus and reproducing (Biedermann, Klepzig & Taborsky, 2009). Within a few weeks females emerge, fly to new trees, and perpetuate the infestation (Rudinsky, 1962). Most trees appear to die from the fungal infection in their tissues (Freeman et al., 2013). Beyond this life cycle, little is known about the effects of any ambrosia beetle in natural habitats because the emphasis of research investigations has been on their presence as pests of commercial agriculture and lumber rather than in natural areas (Wood, 1982; Hulcr & Dunn, 2011).

In late summer 2015, the KSHB was found in the riparian forests of the Tijuana River Valley, making these forests the first natural habitats in California to be substantially attacked by an invasive ambrosia beetle. These forests are dominated by two willow species, Salix lasiolepis Benth. (arroyo willow) and Salix gooddingii C.R. Ball (Goodding’s black willow), which account for more than 80% of the individuals and create the vertical structure of the forest (Boland, 2014a). Both willows were attacked by the KSHB and, within only a few months, tens of thousands of trees were visibly infested. The forests were so obviously negatively impacted by the beetle that the infestation was covered by the local news media (e.g. Graham, 2016; Smith, 2016).

Here I describe the distribution of the KSHB in the riparian habitats of the Tijuana River Valley during the six-month period after first observation, assess the damage caused by the beetle, and discuss the prospects of the habitats recovering from this unusual damage. The overarching goal of this paper is to alert resource managers to this emerging beetle problem in natural habitats.

Study Site

The Tijuana River Valley (32°33.080′N, 117°4.971′W) in San Diego County, California, is a coastal floodplain of approximately 1,500 ha at the end of a 448,000 ha watershed (Fig. 1). The river is a managed intermittent stream that typically flows strongly in winter and spring and is mostly dry in summer (Boland, 2014a). In 2015, however, the main river channels contained water all summer because of unusual rain storms on May 7, 15, July 18 and September 15. The main river splits into two in the center of the floodplain at Hollister Bridge, and the northern arm carries more of the flows than the southern arm because of extensive sedimentation west of Hollister Street within the southern arm.

Figure 1 Map of the Tijuana River Valley.

The location of the riparian forest and riparian scrub habitats within the Tijuana River Valley.

Riparian forests in the river bed are numerically and structurally dominated by S. lasiolepis and S. gooddingii, and the surrounding riparian scrub is numerically and structurally dominated by the perennial shrub, Baccharis salicifolia (Ruiz & Pav.) Pers. (mule fat; Boland, 2014a). Zonation of these three dominant species across the elevation gradient and the factors that produce their zonation were described in Boland (2014a).

The riparian forest and scrub habitats are preserved within three adjoining parks: the Tijuana River Valley Regional Park, the Border Field State Park, and the Tijuana Slough National Wildlife Refuge. The riparian habitats are relatively undisturbed and support numerous reptile, mammal and bird species, most notably the endangered Vireo bellii pusillus Coues (least Bell’s vireo) for which most of the riparian habitats are designated critical habitats (US Fish and Wildlife Service, 1994).

The beetles causing the damage in the riparian habitats of the Tijuana River Valley during 2015–16 were collected and identified as the KSHB by Dr. Akif Eskalen at University of California Riverside (UCR; Eskalen, 2016). These specimens have been stored in the UCR collection.

Materials & Methods

Infestation and damage rates in the valley

To estimate the extent and magnitude of infestation and damage in the valley, I surveyed the entire valley in a stratified random manner. I divided the valley’s riparian forest and scrub habitats into 29 survey units so that the vegetation within each was relatively homogenous in terms of plant species composition, age and density. Within each unit, an accessible survey point was chosen at random from among several accessible points. The location of each survey point was recorded using a handheld GPS unit (Garmin eTrex Venture HC) and the units were mapped (ArcGIS 10.2.2. and projected in Universal Transverse Mercator). I did four surveys in each unit focused on the dominant plant species, S. lasiolepis and S. gooddingii in the forest units, and B. salicifolia in the scrub units.

First, I examined S. lasiolepis, S. gooddingii and B. salicifolia for evidence of KSHB infestation, using binoculars when necessary. A plant was counted as infested (or attacked) if it had beetle holes, extrusion of sawdust plugs or frass, or gumming out of sap (Figs. 2A–2D). Plants with only weeping were noted but included with the non-infested plants in the final count (Hulcr, 2012). All plants examined were within 50 m of the survey point. Surveys were conducted between November 2, 2015 and January 22, 2016. The rate of infestation (the percent infested) was calculated for each forest unit based on S. lasiolepis and S. gooddingii and for each scrub unit based on B. salicifolia.

Figure 2 Photos of the KSHB in the Tijuana River Valley.

(A) Two beetles at the entrance to a tunnel. (B) Holes in the bark of a sycamore. (C) Extrusion of sawdust plugs indicating active burrowing within an arroyo willow. (D) Gumming out of beetle holes in mule fat. (E) Beetle tunnels and associated fungus (black staining) in a cross section through a black willow trunk. (F) Beetle-infested trunks of arroyo willows snapped by the wind.

Second, I examined S. lasiolepis, S. gooddingii and B. salicifolia for evidence of major plant damage. A plant was counted as damaged if it had a recently-broken trunk or major limb (Fig. 2F). All plants examined were within 50 m of the survey point. These surveys were conducted after the first winter storms, between December 14, 2015 and January 22, 2016. The rate of damage (the percent damaged) was calculated for each forest unit based on S. lasiolepis and S. gooddingii and for each scrub unit based on B. salicifolia.

Third, within the forest units I measured the girth of the S. lasiolepis and S. gooddingii trees at breast height. All plants measured were within 50 m of the survey point. An average of 33 (±7) trees was measured in each unit, and the median value within each unit was determined.

Fourth, I recorded the species composition and density of all plant species. All perennial trees and shrubs taller than 0.5 m were identified and counted within a belt transect (20 m × 2 m = 40 m2) that started at the survey point. Transects were done between November 2, 2015 and January 22, 2016. To estimate the total number of S. lasiolepis and S. gooddingii individuals in each forest unit and the total number of B. salicifolia individuals in each scrub unit, the density of these species in the transect was multiplied by the area of the unit. The numbers of individuals infested and damaged in each unit were then extrapolated by multiplying the total number of individuals by the infestation and damage rates in each unit.

Finally, I assigned a measure of wetness to each unit based on the distance of the survey point from surface water during summer 2015. The measure was used to test whether infestation rate in a site was correlated with the relative wetness of the site.

Infestation rates in the common plant species

I examined all common tree and shrub species for signs of infestation when they were encountered while doing the surveys above. As above, a plant was counted as infested if it had beetle holes, extrusion of sawdust plugs or frass, or gumming out of sap, whereas plants with only weeping were noted but included with the non-infested plants in the final count. In total, 23 species were examined, and the rate of infestation (the percent infested) was calculated for each species.

Results

Infestation and damage rates in the valley

Evidence of KSHB infestation was widespread throughout the valley, being observed in all but four of the 29 survey units (Table 1). Rates of infestation were high (61–100%) in 16 of the 22 forest units (Units 1–14, 19, 22). Most of the S. lasiolepis and S. gooddingii in these units had obvious sawdust and frass coming out of abundant holes indicating active boring of the beetles within. Rates of infestation were low (0–10%) in six of the forest units (Units 15–18, 20–21) and in all of the scrub units (0–33%; Units 23–29). Overall, an estimated 287,620 willow trees (71% of the total) and 16,641 mule fat shrubs (4%) were infested by the KSHB (Table 1).

Table 1 Infestation and damage rates within the survey units.

Description of the survey units (stand age, unit area, median willow girth, plant density and estimated total number of plants) and the rates of KSHB infestation and major plant damage (n = number of individuals examined). The total numbers of plants infested and damaged in each unit are extrapolated. In the forest units, data are for S. lasiolepis and S. gooddingii; in the scrub units, data are for B. salicifolia.

Unit	Age	Area	Girth	Density	Infested	Damaged	
				
#	yr	ha	cm	#/40 m2	Total	n	Rate	Total	n	Rate	Total	
A. Riparian forests										
1	35	14.64	130.8	2	7,319	39	74%	5,442	50	12%	878	
2	5	1.78	25.9	36	16,026	143	94%	15,130	67	51%	8,133	
3	15	3.02	52.7	7	5,517	79	100%	5,517	50	98%	5,407	
4	35	5.13	73.7	1	1,282	32	91%	1,162	38	82%	1,046	
5	35	18.08	ND	3	13,561	26	96%	13,040	43	84%	11,354	
6	22	12.26	83.8	3	9,194	37	95%	8,697	44	86%	7,940	
7	10	0.80	39.0	22	4,407	81	100%	4,407	65	94%	4,136	
8	5	2.11	18.3	72	37,953	163	87%	33,063	90	23%	8,856	
9	22	10.21	ND	4	10,211	40	100%	10,211	51	100%	10,211	
10	22	23.04	66.0	3	17,280	42	98%	16,868	68	78%	13,468	
11	35	4.68	81.3	8	9,365	31	100%	9,365	86	78%	7,296	
12	22	3.14	73.7	12	9,421	42	100%	9,421	54	89%	8,375	
13	22	15.01	54.6	10	37,526	70	97%	36,454	52	58%	21,650	
14	22	17.84	34.3	19	84,717	141	75%	63,688	127	17%	14,008	
15	35	18.52	71.1	4	16,204	50	8%	1,296	93	12%	1,917	
16	35	20.75	79.5	5	25,936	35	6%	1,441	66	14%	3,537	
17	35	21.42	83.8	3	16,069	43	0%	0	106	12%	1,971	
18	35	7.06	67.3	4	7,062	44	2%	161	64	28%	1,986	
19	35	6.82	79.8	5	8,524	33	61%	5,166	67	16%	1,399	
20	35	12.86	82.6	3	9,643	40	10%	964	58	5%	499	
21	35	9.56	77.0	3	7,172	35	6%	410	73	7%	491	
22	10	12.83	43.9	15	48,124	60	95%	45,718	82	13%	6,456	
Total		241.6			402,513			287,620			141,011	
B. Riparian scrub										
23	35	76.6	ND	1	19,150	40	33%	6,319	30	0%	0	
24	35	38.0	ND	2	19,016	55	25%	4,754	57	5%	951	
25	15	39.6	ND	1	9,909	34	24%	2,378	42	0%	0	
26	25	31.7	ND	14	110,961	91	0%	0	81	0%	0	
27	35	106.3	ND	6	159,467	55	2%	3,189	50	2%	3,189	
28	35	68.7	ND	6	102,982	50	0%	0	56	0%	0	
29	35	57.8	ND	3	43,378	50	0%	0	52	0%	0	
Total		833.9			464,863			16,641			4,140	
Notes.

ND no data

The forest units with the highest infestation rates were scattered throughout the valley, and the forest units with the lowest infestation rates were contiguous in the center of the valley (Fig. 3). To find a possible cause for this distribution pattern, I looked for correlations between the characteristics of the forest units and their infestation rates. There were no significant correlations between the infestation rates of the forest units and the age of the willow stands (r2 = 0.355, p > 0.05, n = 22), average willow densities (r2 = 0.078, p > 0.05, n = 22) and median willow girths (r2 = 0.1157, p > 0.05, n = 20). However, there was a significant negative correlation between infestation rates of the forest units and distance from surface water (r2 = 0.639, p < 0.01, n = 22), as well as a significant negative correlation between infestation rates of all units and their distance from surface water (r2 = 0.577, p < 0.01, n = 29; Fig. 4). Units that were wet during spring and summer 2015 had the highest infestation rates, whereas units that were dry and far from surface water had the lowest infestation rates. The driest forests were all along the southern arm of the river, which did not receive abundant surface flows during 2015 because of heavy sedimentation.

Figure 3 The KSHB in the Tijuana River Valley in 2015–16.

KSHB infestation levels in the riparian survey units in the Tijuana River Valley. Numbers indicate the 29 survey units and survey points.

Figure 4 Infestation rates vs. wetness.

KSHB infestation rates of survey units as a function of the distance from surface water. The linear regression line, equation and correlation coefficient are for the forest units only (n = 22).

Major plant damage was also widespread in the valley, with some damage observed in all but five of the 29 survey units (Table 1). The magnitude of the damage was particularly high in the eastern forest units (Units 2–13) where more than 50% of the trees had major damage. Before-and-after photos taken of Unit 2 show the kind of damage seen in the more heavily infested and damaged forest units (Fig. 5). The native riparian forest in this unit went from a dense stand of tall willows to a jumble of broken limbs in just a few months. Examination of the broken limbs showed evidence of KSHB infestation and extensive tunneling (Fig. 2E) and showed that the KSHB had so weakened the trunks that the first strong wind was able to snap them (Fig. 2F). The amount of damage within the forest units was significantly positively correlated with the infestation rates in those units (r2 = 0.533, p < 0.01, n = 22); units with high infestation rates generally had high damage rates. Overall, an estimated 141,011 willow trees (35% of the total) and 4,140 mule fat shrubs (1%) were damaged by the KSHB.

Figure 5 Before and after photos.

The forest in Unit 2 during May 2015 and February 2016 showing KSHB-induced damage to the dominant willow trees.

Infestation rates in the common plant species

Of the 23 species examined, 14 species showed obvious signs of beetle attack (Table 2). Among the native species, the tree species showed relatively high rates of infestation (>20%) and the shrub species relatively low rates of infestation (<20%). The four species with the highest rates of infestation (>50%) were all native riparian trees belonging to the Salicaceae family, i.e., S. lasiolepis, S. gooddingii, S. laevigata Bebb. and Populus fremontii S. Watson. Among the non-native species, the three species considered to be the worst invasive plants in the valley, Ricinus communis L., Tamarix ramosissima Ledeb., and Arundo donax L., had relatively low rates of infestation—13%, 3% and 0%, respectively. Young, immature plants of S. gooddingii, B. salicifolia and R. communis showed no signs of infestation although all three species were frequently attacked as adults (Table 2).

Table 2 Infestation rates in the common plant species.

KSHB infestation rates in the common riparian plant species in the Tijuana River Valley.

Species	Common name	Family	# examined	# infested	% infested	
Native species						
Salix lasiolepis Benth.	Arroyo willow	Salicaceae	539	442	82%	
Salix gooddingii C.R. Ball	Black willow	Salicaceae	670	499	74%	
Salix laevigata Bebb	Red willow	Salicaceae	14	9	64%	
Populus fremontii S. Watson	Western cottonwood	Salicaceae	53	28	53%	
Platanus racemosa Nutt.	California sycamore	Platanaceae	28	6	21%	
Baccharis salicifolia (Ruiz & Pav.) Pers.	Mule fat	Asteraceae	486	76	16%	
Baccharis pilularis DC.	Coyote brush	Asteraceae	92	5	5%	
Ambrosia monogyra*	Singlewhorl burrobrush	Asteraceae	33	1	3%	
Salix exigua Nutt	Narrow-leaf willow	Salicaceae	88	1	1%	
Malosma laurina (Nutt.) Nutt. Ex Abrams	Laurel sumac	Anacardiaceae	7	0	0%	
Peritoma arborea (Nutt.) H.H. Iltis	Bladderpod	Cleomaceae	9	0	0%	
Sambucus nigra L.	Blue elderberry	Adoxaceae	31	0	0%	
Scirpus spp.	Tule	Cyperaceae	5	0	0%	
Artemisia dracunculus L.	Tarragon	Asteraceae	19	0	0%	
Chloracantha spinosa (Benth.) G.L. Nesom	Spiny aster	Asteraceae	2	0	0%	
Isocoma vernonioides (Nutt.) G.L. Nesom	Coastal goldenbush	Asteraceae	14	0	0%	
Non-native species						
Schinus terebinthifolius Raddi	Brazilian pepper	Anacardiaceae	15	7	47%	
Nicotiana glauca Graham	Tree tobacco	Solanaceae	40	8	20%	
Ricinus communis L.	Castor bean	Euphorbiaceae	123	16	13%	
Eucalyptus spp.	Gum tree	Myrtaceae	21	2	10%	
Tamarix ramosissima Ledeb.	Salt cedar	Tamaricaceae	32	1	3%	
Tropaeolum majus L.	Garden nasturtium	Tropaeolaceae	5	0	0%	
Araujia sericifera Brot.	Cruel vine	Apocynaceae	5	0	0%	
Phytolacca icosandra L.	Tropical pokeweed	Phytolaccaceae	5	0	0%	
Schinus molle L.	Peruvian pepper	Anacardiaceae	7	0	0%	
Myoporum laetum G. Forst.	Myoporum	Scrophulariaceae	16	0	0%	
Acacia cyclops G. Don	Cyclops wattle	Fabaceae	2	0	0%	
Arundo donax L.	Giant reed	Poaceae	42	0	0%	
Immature plants (<1 year old)						
Salix gooddingii	Black willow	Salicaceae	72	0	0%	
Baccharis salicifolia	Mule fat	Asteraceae	30	0	0%	
Ricinus communis	Castor bean	Euphorbiaceae	32	0	0%	
Notes.

* (Torr. & A. Gray) Strother & B.G. Baldwin.

Discussion

The Tijuana River Valley is the first natural riparian habitat in California to be substantially attacked by an invasive, non-native ambrosia beetle; KSHB attacks in California have previously been observed only in agricultural and urban settings (Eskalen et al., 2013). The goals of this study were to describe the KSHB infestation in the Tijuana River Valley and to alert resource and land managers to the beetle problem in natural habitats. Several findings from this study have significance for resource and land managers.

First, the KSHB attack caused extensive damage to the trees in the riparian forests. Many dense stands of tall willows were reduced to broken trunks, making the magnitude and severity of the beetle attack on a par with a medium-severity fire (Brumby et al., 2001). Furthermore, the damage appeared to occur rapidly—in just a matter of months. The KSHB was first observed in the valley in August 2015 (J Boland, pers. obs., 2015) and as many as 100,000 trees had their branches or trunks snapped by the first winter wind storms in December 2015. It is likely that the beetle was present in low numbers well before it was first observed and that it had time to build up its population to outbreak levels (Crooks, 2005). However the apparent speed of attack means that if resource managers are to control the spread of the beetle in their riparian forests, they will need to develop an early detection program that will discover the beetle when it is at very low population levels (Porter, 2007). They will also need a rapid response program in place so that they can respond as soon as a small population of the beetle is discovered. A program currently being discussed in San Diego County involves the monitoring of apparently uninfested riparian habitats and the removal of any infested trees, followed by their chipping or solarization (placing in the sun under a clear tarp for several months; Jones & Paine, 2015). A research group at University of California Riverside is exploring many control options, including fungicides, pesticides and biocontrol, primarily for the avocado industry but their findings are likely to assist in the control of the beetles in native habitats as well (Spann, 2015; Eskalen, 2016).

Second, KSHB infestation rates in the forest units were significantly correlated with the proximity to surface water or the wetness of the unit. This correlation suggests that the beetles were more attracted to or active in the wetter parts of the valley where trees grow in or near water. In these wetter areas, trees would have high internal moisture content, because the moisture content of plant tissues is positively correlated with that of the surrounding soil (Constantz & Murphy, 1990), and they would be growing vigorously, because the water and soils are nutrient enriched due to frequent cross-border sewage flows. The idea that an ambrosia beetle would prefer to attack trees growing in wet, nutrient-rich conditions has support in the literature; Rudinsky (1962), in his review of the ecology of the Scolytidae, reported that growth of the fungi and larvae of ambrosia beetles was positively correlated with moisture content of the host plants, and Coyle, Booth & Wallace (2005) found that a clone of the eastern cottonwood, Populus deltoides Bartram, was attacked by ambrosia beetles significantly more frequently when the trees received liquid fertilizer than when they received control treatments. The factors governing KSHB infestation have yet to be fully understood, but the observed correlation with wetness in the Tijuana River Valley suggests that those searching for KSHB infestations in other parts of southern California should focus especially on Salicaceae-dominated forests near nutrient-enriched water.

Third, KSHB impacts appear to be a novel form of disturbance. Riparian species in California are adapted to many kinds of disturbances, including floods, fires, avalanches, debris flows, ice scours, windstorms and droughts (Pettit & Naiman, 2007). Such disturbances may kill many plants but, importantly, they usually leave large open expanses of sandy sediments and some living trees. The riparian vegetation reestablishes quickly after such events because new seedlings can establish abundantly in the open, sandy areas and the living trees can grow back from stump sprouts (Rood et al., 2007; Wintle & Kirkpatrick, 2007; Boland, 2014b). The KSHB, however, has disturbed the forest in a unique way; the damaged forests are littered with woody debris that has fallen from the canopy so there are few patches of open sediment for seedlings, and there appear to be few living tree stumps from which sprouts can grow. Opportunities for revegetation via seedlings and resprouts therefore appear to be greatly limited. Whether the riparian vegetation can respond to this unique disturbance caused by the KSHB remains to be seen. Because the KSHB was not observed to attack young willow and mule fat plants, removal of woody debris in sites suitable for plant recruitment might be a good strategy for promoting the recovery of damaged riparian habitats.

Fourth, the KSHB attack appears to be having ecosystem-wide impacts. The KSHB showed a preference for the willows S. lasiolepis and S. gooddingii, the dominant trees in the Tijuana River Valley (Boland, 2014a). Reduction or loss of these foundation species will drastically alter the canopy architecture, understory microclimate, and ecosystem processes such as productivity and water balance within the forest units (Ellison et al., 2005). Three species that are considered to be among the worst invasive species in California (Cal-IPC, 2006), R. communis, T. ramosissima and A. donax, are likely to thrive in the damaged forests. All three species were present but relatively inconspicuous in the forest prior to the beetle infestation, and all are likely to grow tall and thrive in the absence of a willow canopy (J Boland, pers. obs., 2010). A new riparian community composed of invasive shrubs and herbaceous weeds is unlikely to provide the same habitat quality and food-chain support for animal species in the valley, including the endangered Vireo bellii pusillus. It is therefore possible that the KSHB is affecting not only its preferred host species but also the structure and function of the entire ecosystem. Managers will need to monitor many aspects of the damaged forests and pay particular attention to the spread of non-native plant species in these new forest habitats.

Supplemental Information

Data S1 Complete wetness data

Distance from surface water data and regression lines showing regression statistics.

Click here for additional data file.

I thank Deborah Woodward for assistance in the field, helpful discussions, and comments on drafts of this manuscript, Monica Almeida for producing the maps, and Jeff Crooks, Mike Picker, Dezene Huber, Leland Humble, Richard Stouthamer and an anonymous reviewer for their valuable comments on an early draft. I also thank the US Fish and Wildlife Service, US Navy, California State Parks, and the San Diego County Parks and Recreation Department for preserving the Tijuana River Valley riparian habitats and making them available for study.

Additional Information and Declarations

Competing Interests

Author Contributions

Data Availability

The author declares there are no competing interests.

John M. Boland conceived and designed the experiments, performed the experiments, analyzed the data, contributed reagents/materials/analysis tools, wrote the paper, prepared figures and/or tables, reviewed drafts of the paper.

The following information was supplied regarding data availability:

The raw data has been supplied as Data S1.

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
