# Peer review of "The impact of an invasive ambrosia beetle on the riparian habitats of the Tijuana River Valley, California"

_PeerJ, doi:10.7717/peerj.2141_

## Round 0.1 · original submission · Minor Revisions

This MS is generally well-written and provides a good assessment of a fairly newly introduced ambrosia beetle in California. Early assessments, like this, are an important part of the natural historical record. The reviewers all have given the paper favorable reviews, and I concur with their impressions and opinions.

Two notes beyond what the reviewers mentioned:

1. I would suggest standardizing the abstract by removing the within-abstract subheadings. I understand the author's rationale here, but it is a bit extraneous.

2. Please consider removing bold highlighting within the text. E.g. lines 147, 154, 160, 221, 237, etc. Anywhere that the bolding is not a bonafide heading or subheading, it could be removed and not detract from the impact of the paper.

Please respond with a revised MS, seriously considering the excellent suggestions and advice from the reviewers.

Thank you for your submission to PeerJ.

Reviewer 1 ·

Basic reporting

No comments

Experimental design

No comments

Validity of the findings

No comments

Comments for the author

This ms describes a relatively simple and straightforward study concerning colonization of a riparian forest by Kuroshio shot hole borer, an exotic invasive species. To date, little has been published on the Kuroshio shot hole borer since its introduction into southern California several years ago. I recommend accept after minor revisions, and offer the following suggestions:

1) Ln. 1: Delete “devastating”
2) Ln. 45-47: Revise to “Infestation of KSHB in the riparian forests of the Tijuana River Valley has led to a drastic alteration of structure.”
3) Ln. 59: This is not accurate – revise.
4) Ln. 60: Define “destroying”
5) Ln. 139-140: How was infestation by KSHB distinguished from that of PSHB? Is this possible given the techniques used, and if not, is it appropriate to attribute all activity to KSHB? Elaborate.
6) Ln. 221: Replace “devastated” with a more appropriate and descriptive term.
7) Ln. 223 (and throughout ms): Replace “destruction” with “change in structure”. In general, the ms contains many terms that assign values to effects (i.e., without fully defining them). I find this inappropriate and distracting. Review and revise (as appropriate) throughout the ms.
8) Ln. 233: Define “solarization” as many readers may not be familiar with the term.
9) Ln. 260: Delete “in spring”
10) Ln. 296: Replace “K.R.” with “K.D.”

·

Basic reporting

Basic Reporting
Please be consistent I providing the authority for all scientific names listed in the paper. Authorities given for Salix spp. (lines 88-89) and Baccharis (line 114) but not for avocado (line 72), Populus fremonti, Ricinus communis, Tamarix ramosissima and Arundo donax (lines 206-208) and other plant species listed in Table 2; insect species (lines 58, 60 and 66) and birds (line 121).
Balsam woolly adelgid reference somewhat dated and appears to have been selected to demonstrate impacts on avian communities; Hemlock woolly adelgid (Adelges tsugae) or emerald ash borer (Agilus planipennis) are more recent examples of introduced pests impacting riparian communities.
Common names used in this paper are not common names accepted by the Entomological Society of America but have been proposed to distinguish the two undescribed species discussed on lines 64-72 from Euwallacea fornicatus (Eichhoff) . The author should clarify the second paragraph of the introduction so that the reader is aware that PSHB and KSHB are part of a species complex distinguishable from each other by sequence divergence in mtDNA that includes Euwallacea fornicatus. The species of Fusarium vectored differs between species
Please correct the specific epithet for E. fornicatus on line 66.
Would prefer that attack of tree species by ambrosia beetles be referred to as infested rather than an infected (line 92 and throughout the manuscript and accompanying tables). If one is referring to fungal growth in galleries in the wood it could be referred to as an infected. Survey done was for visual evidence of attack or infestation; it did not determine if the hosts were infected by Fusarium.
• Line 127- Infestation and damage rates in the valley
• Line 138 - …evidence of KSHB infestation or …. evidence of KSHB attack
• Line 141 - unattacked plants
• Line 143 – A rate of attack (the percent infested)
• Line 161 - ….whether the attack rate at……..
• Line 165 -170 change infected to infested or attacked e.g. ……..for signs of attack…….
Line 209-210: reword so that it is clear that you are distinguishing between attack rate in mature and immature plants

Experimental design

Methodology used allows for a relatively rapid assessment of damage across plant communities at the study site. Distance from water may be a surrogate for habitat dryness but it is not a reliable measure of wood moisture content and thereby a measure of suitability of wood for ambrosial fungal growth.

Validity of the findings

Conclusion that KSHB impacts differed along a gradient of dryness is supported by the data presented for the riparian forest, however the low levels of attack observed in mule fat could also be the result of poor host suitability for the ambrosial fungus (Fusarium) required for larval development in the galleries. As host suitability for fungal growth was not determined, one cannot conclude that habitat dryness was the cause of the responses noted.
Suggestion that tree growers could decrease the watering of the trees to reduce the risk of KSHB attack should be removed from the paper as many factors including drought stress have been identified that impact levels of ambrosia beetle attack (see Oliver and Mannion 2001 Environ. Entomol. 30(5): 909-918; Coyle et al 2005 J. Econ. Entomol. 98(6): 2049-2057). Experimental trials suggested need to be completed before one could make such a recommendation.
KHSB impacts appear to be a novel disturbance. Observation that KHSB did not attack young willow and mule fat plants could alternatively be explained by a requirement for a minimum host diameter for gallery construction by the founding females. Were attacks evident in the smaller diameter twigs of mature willows and mule fat of a diameter equivalent to young plants?

Comments for the author

You are to be commended on completing such a rapid assessment of impacts of an invasive ambrosia beetle in this riparian habitat so soon after its initial discovery. I would also suggest that you review the following reference Kasson et all 2013 in Fungal Genetics and Biology 56:147-157 as it provides some insights into the cryptic nature of the causal agents both Euwallacea spp and Fusarium spp. I would also recommend that at a minimum specimens of Euwallacea be collected and deposited in an accessible reference collection to ensure that material is available for the eventual description of this taxon.

·

Basic reporting

This is an important and timely contribution that documents the devastation caused to a native riparian vegetation by an invasive shot hole borer. The author has a clear survey protocol and gives clear results showing 1. The level of devastation as a function of the different host plant species, 2. The relative dryness of the plots as indicated by the distance from open water. This work is showing first of all the rapidity with which the beetles can cause the complete devastation of an important habitat. The recommendation offered by the author with regards to what managers of such areas can do is suitable unfortunately it may be really difficult to detect infestations in the early phase.
Some minor editorial issues: your word processor likes to change fornicatus into fornicates
Line 72 I think it is more appropriate there to cite Eskalen et al 2013 maybe with an additional statement that a current list of reproductive hosts is given on the website (Eskalen 2016)
In the table 1. I think we need some clarification in the heading showing the infected and damaged individuals I assume that in each of these the first column indicates the number or infected/damaged plants per observation point consisting of a circle with a 50 m radius. Please make that clearer and also that the total of number of plants infected/damages is an extrapolation of the rate times the total number of plants found in that survey unit.

Experimental design

The design is sufficient for the descriptive nature of the work. The intend of this paper is a description of the devastation caused by this invasion. The methods employed are sufficient to show what the author wished to convey

Validity of the findings

Again this is a descriptive paper which clearly shows the destruction caused by this invasion. As such this is an important paper because shows for the first time how bad these beetles species can affect an important habitat.

---

## Round 0.2 · accepted · Accept

The author has responded in a comprehensive fashion to the comments and concerns of the three reviewers. This MS is now acceptable for publication in PeerJ.

As always, I would like to recommend to the author that the reviewers' comments be made public alongside the published MS.